# Learning Disentangled Representations for Fairness with Limited Demographics

## Abstract

Fair representation learning is a promising way to mitigate discrimination in downstream tasks. Many existing fair representation learning methods require access to sensitive information, but the collection of sensitive information is often difficult and even involves privacy issues. Additionally, a model trained to be fair with respect to one sensitive attribute may not ensure fairness for other sensitive groups. Thus, how to flexibly address fairness issues when we have limited access to sensitive information is a challenging problem. In this work, we answer this question: "**given limited sensitive information, can we learn a representation to be fair w.r.t. varying sensitive groups?**" To achieve this, we propose a novel two-step framework. We first learn a disentangled representation by employing Non-linear Independent Component Analysis (Nonlinear ICA). Second, we remove sensitive information in the latent space to obtain fair representation. The learned representation can be easily adapted to be fair w.r.t different sensitive groups and to be used for different downstream tasks without re-training. Among the entire process, only a small portion of sensitive information is required in the second step to learn a fair representation. We compare with methods that require different amounts of sensitive information on real-world image and tabular datasets. We empirically demonstrate the utility and flexibility of our approach, and our method is capable of achieving improved fairness results in various tasks.

## 1 Introduction

Fairness in machine learning is a vital issue because more and more machine learning algorithms are applied to high-risk tasks, such as loan granting, medical applications, and university admission (Corbett-Davies & Goel, 2018). In many cases, machine learning models inevitably encode human discrimination due to the distribution of data.

Currently, fair representation learning has been proven to be a good way to address fairness concerns (Zemel et al., 2013). Specifically, a fair representation encodes the input features in the latent space and reduces the impact from sensitive groups (Liu et al., 2022; Quadrianto et al., 2019; Kehrenberg et al., 2020; Park et al., 2021). Various approaches have been explored, including adversarial training (Edwards & Storkey, 2015; Madras et al., 2018), regularization (Tran et al., 2021; Sarhan et al., 2020; Quadrianto et al., 2019), reweighting (Krasanakis et al., 2018; Yan et al., 2022; Li & Liu, 2022), and disentanglement (Park et al., 2021; Locatello et al., 2019a; Sarhan et al., 2020), and distribution mapping (Balunovic et al., 2021; Cerrato et al., 2022)

One problem with many existing methods is that they require all sensitive information when learning a fair representation. Collecting sensitive information is expensive in the real world and even raises privacy concerns. This limits the application of many of the existing methods in practice.

Two strategies have been proposed to tackle this challenge: methods that require no sensitive information and methods that require a part of sensitive information. Distributionally Robust Optimization (DRO) (Hashimoto et al., 2018) and Adversarial Representation Learning (ARL) (Lahoti et al., 2020), are designed to function without accessing any sensitive information. These approaches focus on optimizing the worst-performing sample subsets to improve group fairness. However, as the authors point out, such an approach may get stuck in optimizing noisy data, leading to a significant drop in performance (Hashimoto et al., 2018). Moreover, it introduces a strong constraint on the objective function by only optimizing the worst-performing samples. Chai et al. (2022) introduced a

novel methodology employing a knowledge distillation framework to mitigate fairness issues without relying on sensitive information. However, this strategy requires a larger teacher network to overfit the training set, resulting in parameter inefficiency. Grari et al. (2021) proposed a method utilizing a Bayesian variational autoencoder to infer sensitive information. It demands substantial expert knowledge of the data. Moreover, the effectiveness of this method significantly depends upon the accuracy of the sensitive attribute proxies.

The alternative approach leverages partial sensitive information, it can be categorized into two groups: optimization with partial dataset and pseudo-labeling-based methods. Fair models with Bilevel Optimization (BiFair) (Ozdayi et al., 2021), and Just Train Twice (JTT) (Liu et al., 2021) are two exemplar methods of the first category. BiFair assigns a weighted training set to achieve utility and fairness, where the weight of each training sample is learned through a partially sensitive labeled dataset. JTT optimizes the worst performance samples by retraining on up-weighted training samples. They held out a validation set to tune the up-weight factor. However, BiFair relies on bilevel optimization, which is inherently challenging to resolve effectively (Beck et al., 2023). JTT needs training twice, which can become prohibitively expensive for large-scale training, and the performance of JTT is sensitive to hyperparameters (Liu et al., 2021). Another type of approach is through pseudo-labeling (Nam et al., 2022; Jung et al., 2022). These methods first train a network with a limited amount of sensitive information and subsequently assign pseudo-sensitive labels to the dataset. Then they use robust training techniques to ensure equitable performance across different sensitive groups. However, the accuracy of pseudo-sensitive labeling significantly influences overall performance. It poses considerable risks in high-stakes tasks, where errors in labeling could lead to substantial consequences.

Additionally, another challenge is that representations lack the flexibility to adapt fairness across different attributes. For example, while current methods enable the learning of representations that are fair for gender, these cannot be easily modified to achieve fairness with respect to race. Retraining such representations is often expensive.

Creager et al. (2019) first proposed the concept of flexible representation learning. Subsequent studies (Usama & Chang, 2022; Kehrenberg et al., 2020) extended this idea by advocating for the learning of representations that can accommodate various sensitive attributes. They enumerate all potential sensitive attributes as candidates and subsequently learn two vectors to separately encode non-sensitive and sensitive information. The non-sensitive vector is then employed to train a fair classifier. Although this method has improved flexibility, it is necessary first to clarify all possible sensitive groups. Retraining the representation may still be required if the target sensitive information is not considered from the outset.

In summary, existing methodologies that address limited sensitive information encounter optimization challenges or risks for predicting sensitive attributes. Moreover, they often lack the necessary flexibility. While there are methods designed to enhance flexibility, they need all sensitive attributes during training. This requirement not only imposes a significant burden on collecting sensitive information but also necessitates re-training whenever a new sensitive attribute emerges.

In this work, we propose a new solution to address the above-mentioned challenges in terms of flexibility and ease of use in fairness representation learning. We utilize a generative disentangled representation learning method to map the input into each independent latent space. The learning process is guided by Nonlinear Independent Component Analysis (Nonlinear ICA), which is theoretically identifiable—capable of uncovering the true generative variables. We postpone the discussion of how Nonlinear ICA differs from other disentanglement learning approaches to Section 3.1. After learning the disentangled representation, we identify dimensions correlated with sensitive information by calculating differences in conditional distributions. This disentangled representation can then adapt to meet fairness requirements across various sensitive groups. Notably, the procedure for learning fair representations requires only a small portion of sensitive information. Furthermore, the generative nature of the latent vectors allows for straightforward manipulation, facilitating image modification and generation with altered labels.

We summarize our contributions as follows:

- Our method learns a flexible and fair representation, without presetting sensitive information. Thus, there is no need to retrain the representation and it can be easily adapted to be fair with varying sensitive groups.

- Only limited sensitive information is required, which protects sensitive data.
- Our method is versatile, enabling a range of downstream tasks including fair classification and image generation.
- Experimental results validate that our method is comparable and even outperforms state-of-the-art methods on image and tabular datasets.

## 2  RELATED WORK

**Fairness with limited demographics.** Several approaches have been proposed to mitigate bias when sensitive information is limited. BiFair (Ozdayi et al., 2021) dynamically adjusts weights for data points to achieve both good utility and low bias for the model. Just Train Twice (JTT (Liu et al., 2021)) first trains an ERM model, then trains with unweighted error data points to boost performance in the worst-performing group. Spread Spurious Attribute (SSA (Nam et al., 2022)) utilizes partially labeled sensitive information to infer sensitive labels in the training set, subsequently employing this pseudo-sensitive information to train a robust model. Similarly, Confidence-based Group Label assignment (CGL (Jung et al., 2022)) uses pseudo-labeling but differs in its approach. A sensitive label classifier is trained to assign pseudo-sensitive labels based on confidence levels. CGL also incorporates a robust training method to mitigate bias in the model. FairLisa (Liu et al., 2024) utilizes an adversarial debiasing approach, sequentially training a debiasing filter and discriminator. However, a recent study (Han et al., 2023) shows that "adversarial debiasing methods exhibit instability." In contrast to existing methods, our approach avoids adversarial training, thereby enhancing stability. Moreover, our learned representations can be adapted to promote fairness for various sensitive attributes without necessitating re-training.

**Flexible fair representation learning.** Flexible fair representation learning is a methodology designed to ensure fairness across various sensitive groups at test time, as first introduced by (Creager et al., 2019). They utilized a variational autoencoder to segregate non-sensitive and sensitive information into distinct latent vectors. The sensitive latent vector is specifically designed to encode diverse sensitive attributes, enhancing the model's flexibility. These vectors are trained to ensure minimal mutual information. Subsequently, the non-sensitive latent vector is employed to train fair classifiers. Aligning with this concept, Usama & Chang (2022) employed Maximum Mean Discrepancy (MMD) in VAE to ascertain the independence of the non-sensitive latent vector from the sensitive attribute. Similarly, Kehrenberg et al. (2020) proposed a flow-based model connected with an adversarial mechanism to reduce sensitive information leakage during encoding. Our work differs from these methodologies in two critical aspects: (1) Existing methods necessitate the enumeration of all sensitive attributes during training. In contrast, our approach eliminates this requirement, offering a significant advantage in adaptability. (2) Creager et al. (2019) and Usama & Chang (2022) require the availability of all sensitive attributes during training, and Kehrenberg et al. (2020) requires a training set where the mutual information between the target label and sensitive information is minimized. Our method efficiently works with a limited subset of sensitive labels, enhancing flexibility and significantly strengthening privacy protections.

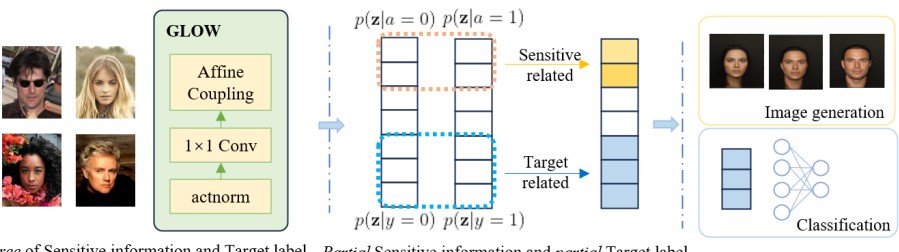

Figure 1: Overview of our learning framework. Our method consists of three steps: In the first step, we use the GLOW model to learn a disentangled latent representation. In the second step, we employ a small portion of samples labeled with sensitive and target information to identify the dimensions of the latent representation that correlate with sensitive information and target labels, respectively. In the third step, we employ the learned representation for performing different downstream tasks.

## 3 METHOD

The objective of our work is to learn a fair and disentangled representation. Non-linear Independent Component Analysis (nonlinear ICA) is widely used in the field of representation learning due to its theoretical guarantees of identifiability, which ensures the uniqueness of the representation (Hyvärinen et al., 2023). We leverage the principles of nonlinear ICA and expand them within the realm of fair representation learning. Practically, the flow-based model, as an invertible neural network, aligns perfectly with the requirements of nonlinear ICA. In this section, we first introduce the preliminaries of our work, followed by presenting our approach.

### 3.1 PRELIMINARIES

**Nonlinear ICA in disentanglement** The objective of nonlinear ICA is to uncover the hidden generative processes that influence the observed data and to separate mixed signals into their independent sources. The nonlinear ICA decomposes an observed variable into a mixing function and independent components. Specifically, we assume that an observed data $\mathbf{x}$ is generated by a generating latent vector $\mathbf{z}_{gen}$ through a mixing function $f$, where $\mathbf{z}_{gen}$ is unobservable. Nonlinear ICA estimates $\mathbf{z}_{gen}$ by learning an invertible mixing function $f$ from $\mathbf{x}$, such that the learned representation $\mathbf{z} = f^{-1}(\mathbf{x})$ is close to $\mathbf{z}_{gen}$. The mixing function $f$ maps the latent representation $\mathbf{z}$ to the input observed data $\mathbf{x}$: $f : \mathbf{z} \mapsto \mathbf{x}$, and since $f$ is invertible, we also have: $f^{-1} : \mathbf{x} \mapsto \mathbf{z}$.

Hyvarinen et al. (2019) and Khemakhem et al. (2020) give a theoretical analysis that by providing some auxiliary variables, the nonlinear ICA is able to find a latent variable up to a transformation of the generating latent variable. Auxiliary variables can be time structures, class labels, or other variables that latent variables are conditioned on. Sorrenson et al. (2020) further demonstrates that the representation estimated by a flow-based model can be a translation and scaling of the generating latent vector.

**Comparing other Disentanglement method**. Extensive research has been conducted on learning disentangled representations, utilizing methods such as probabilistic component analysis (Tipping & Bishop, 1999) and variational autoencoders (VAEs) (Kingma, 2013). The majority of these studies concentrate on VAEs and employ regularized objectives to direct the learning process towards disentangled latent representations (Khemakhem, 2022). Unfortunately, these methods lack identifiability guarantees. An identifiable method can uniquely estimate and reveal the true latent variables. The absence of identifiability results in arbitrary representation, where the representation may alias the true latent variable (Hyvärinen et al., 2023). Consequently, the learned latent representation might couple sensitive information with target information, posing significant challenges for debiased learning. Additionally, Locatello et al. (2019b) demonstrate that without guaranteed identifiability, the disentanglement of latent representations is highly sensitive to random seeds, making it less suitable for downstream tasks. Fortunately, nonlinear ICA provides a theoretical foundation that ensures identifiability (Mathieu et al., 2019; Locatello et al., 2019b; Rolinek et al., 2019). Formally, **Theorem 1** in (Hyvarinen et al., 2019) states that under the assumptions of invertibility for the mixing function, existing of auxiliary variable, and smoothness for the density function of $\mathbf{z}$, the function estimated by nonlinear ICA yields a *consistent* estimator of the demixing function. This theorem suggests the applicability of the identifiability property of nonlinear ICA to recover the generative latent variables $\mathbf{z}_{gen}$.

**Flow-based model implements nonlinear ICA.** A flow-based model is a type of generative model, which uses an invertible neural network (INN) to transform the input probability distribution into an easy-to-sample probability distribution. Concretely, the input data $\mathbf{x}$ is generated from the latent representation $\mathbf{z}$ through the function $f$. The function $f$ is a composite function consisting of several simple and invertible functions, $f = f_1 \circ ... \circ f_k$, where $k$ is the number of composite functions. By dedicated design, the inverse of $f$ and the log-determinant of the Jacobian matrix can be easily computed (Dinh et al., 2014; 2016; Kingma & Dhariwal, 2018). For a flow-based model, the relationship $\mathbf{z} = f^{-1}(\mathbf{x})$ aligns with the setting of nonlinear ICA. Moreover, due to its inherent nonlinearity and invertibility, the flow-based model can be a natural choice for constructing the mixing function in nonlinear ICA.

## 3.2 SENSITIVE-INFORMATION-FREE DISENTANGLED REPRESENTATION LEARNING

Figure 1 provides an overview of our learning framework. In this work, we use the following notations: $\mathbf{x} \in \mathcal{X}$ as the input, $a \in \mathcal{A}$ as the sensitive information, $y \in \mathcal{Y}$ as the target label, $\mathbf{z} \in \mathbb{R}^n$ as the latent representation, and $\mathbf{u} \in \mathbb{R}^u$ as an auxiliary variable.

To achieve fair representation, our approach employs nonlinear ICA to learn disentangled representations. To guarantee disentanglement in $\mathbf{z}$, the distribution of $\mathbf{z}$ should be factorial. And to guarantee that the estimated $\mathbf{z}$ is a simple transformation of $\mathbf{z}_{gen}$, the distribution $p(\mathbf{z}|\mathbf{u})$ should be different conditioned on different $\mathbf{u}$. By following a simplified and valid assumption in the nonlinear ICA literature that $\mathbf{z}$ follows the multiplication of univariate Gaussian distributions, we formulate the conditional distribution of $\mathbf{z}$ as:

$$p(\mathbf{z}|\mathbf{u}) = \Pi_{i=1}^n \frac{1}{\sqrt{2\pi}\sigma_i(\mathbf{u})} \exp[-\frac{1}{2}(\frac{z_i - \mu_i(\mathbf{u})}{\sigma_i(\mathbf{u})})^2], \tag{1}$$

$\mu_i$ and $\sigma_i$ are the mean and variance in the latent space.

**Choice of $\mathbf{u}$:** The choice of auxiliary variable $\mathbf{u}$ is not restrictive, as Hyvarinen et al. (2019) point out, the auxiliary variable can take various forms such as a class label, patch index, time index, and more. The learned $\mathbf{z}$ varies with different auxiliary variables $\mathbf{u}$. However, the linear relationship between $\mathbf{z}$ and $\mathbf{z}_{gen}$ remains unchanged. To be precise, in a Gaussian latent space, we have $z_{gen}^i = w^i z^j + b^i$ (Sorrenson et al., 2020), where the parameters $w$ and $b$ are functions of $\mathbf{u}$. In fairness consideration, we expect to disentangle sensitive information $a$ in $\mathbf{z}$, which means at least one dimension of $\mathbf{z}$ is not conditioned on $a$. Hence the auxiliary variable $\mathbf{u}$ can be observed variable other than sensitive information $a$. In practice, the selection of $\mathbf{u}$ adheres to several criteria: Firstly, neither the sensitive information $a$ nor the target label $y$ is chosen as $\mathbf{u}$. Second, $\mathbf{u}$ is composed of features with a relatively balanced distribution. Third, to enhance fairness outcomes, variables that are highly correlated with $a$ should be avoided. Based on the criteria, we compute the Pearson correlation coefficient between each annotation and $a = $ male, as well as the data balance for each annotation, where a ratio of 0.5 indicates perfect balance in the CelebA dataset. The results are shown in Figure 2. We select the auxiliary variable by choosing annotations with correlation coefficients close to 0 and balance ratios near 0.5. In practice, we utilized the limited sensitive labeled subset to determine the correlation, selecting $\mathbf{u}$ = {*High Cheekbones*, *Mouth Slightly Open*, *Straight Hair*, *Oval Face*} in the CelebA dataset. We provide an extended discussion in Appendix D.

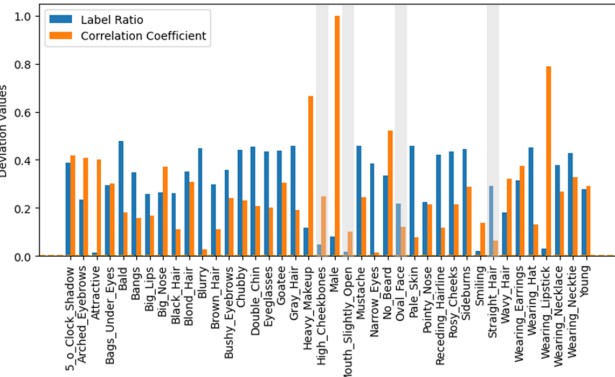

Figure 2: Bar plot of Pearson correlation coefficients and balance ratios in the CelebA dataset. Orange bars denote the absolute value of Pearson correlation coefficients between annotations and 'Male'. Blue bars illustrate the balance ratio for each annotation. Values close to 0 for both orange and blue bars indicate less correlation and higher balance.

By the description of nonlinear ICA in Section 3.1, we learn a mixing function $f$ that maps from the latent representation $\mathbf{z}$ to the observed data $\mathbf{x}$. The training objective is to minimize the negative log-likelihood of $\mathbf{z}$ in the latent space (Sorrenson et al., 2020):

$$L(\theta) = \mathbb{E}_{\mathbf{x},\mathbf{u} \in D}[\frac{1}{n}\sum_{i=1}^n (\frac{(f^{-1}(\mathbf{x};\theta) - \mu_i(\mathbf{u};\theta))^2}{2\sigma_i^2(\mathbf{u};\theta)} + \log(\sigma_i(\mathbf{u};\theta)) + \sum_{j=1}^k \log|\det \frac{\partial f_j^{-1}}{\partial f_{j-1}^{-1}}|)], \tag{2}$$

where $D$ denotes dataset, and $\theta$ is the parameter of the invertible function $f$. The objective function implies that we learn a Gaussian mixture in the latent space, and the number of components of the Gaussian mixture is the same as the number of the distinct auxiliary variable $\mathbf{u}$. In practice, we apply the flow-based model to formulate the invertible transformation function $f$ in equation 2.

### 3.3 Identifying Semantic in disentangled latent representation

We explore dimensions within latent space correlating to sensitive information and target labels, respectively. For the sake of clarity, we detail the process of identifying sensitive information encoded in the latent representation. The process for identifying target labels in the latent space remains the same approach. We compute the correlation between each dimension in the latent representation $\mathbf{z}|\mathbf{u}$ and the sensitive information $a$: We map the input data to the latent space by the function $f^{-1}$ mentioned in section 3.2. We obtain a latent space $\mathcal{Z}$, where each point within this space represents a disentangled latent representation $\mathbf{z}$. We denote $z_i|\mathbf{u}$ as the $i$-th dimension of the conditional latent $\mathbf{z}|\mathbf{u}$, $z_i|a, \mathbf{u}$ is the $i$-th dimension of latent $\mathbf{z}|\mathbf{u}$ conditioned on the sensitive information $a$. If the $i$-th dimension correlates to the sensitive information $a$, it expects a distribution shift according to different $a$. In other words, $p(z_i|a = 0, \mathbf{u}) \neq p(z_i|a = 1, \mathbf{u})$. Hence, we measure the distance of distribution similarity on $p(z_i|a = 0, \mathbf{u})$ and $p(z_i|a = 1, \mathbf{u})$. We define the MMD score function $m$:

$$m(z_i, a|\mathbf{u}) = \mathrm{MMD}(\mathcal{F}, p(\hat{z}_i|a = 0, \mathbf{u}), p(\hat{z}_i|a = 1, \mathbf{u})), \tag{3}$$

where $\hat{z}_i = \frac{z_i}{||z_i||}$ is the normalization of $z_i$, MMD is the maximum mean discrepancy, $\mathcal{F}$ is the Gaussian kernel function. Equation 3 shows the distribution similarity of latent representations between different sensitive groups. Higher $m(z_i, a|\mathbf{u})$ indicates lower similarity between different sensitive groups on the $i$-th latent dimension.

It is worth noting that when searching for sensitive-related dimensions, we do not need to use all the sensitive information in the training set. We empirically justify that only a few samples are needed to search the sensitive-related dimensions in the latent space. We employ Spearman rank correlation coefficient to measure the correlation between two rankings. It is defined as $\rho = 1 - \frac{6||\Delta\mathbf{r}||^2}{n(n^2-1)} \in [-1, 1]$ where $\Delta\mathbf{r}$ is the difference of the two rankings. A $\rho$ value approaching 1 indicates a strong positive correlation between the two rankings. Take CelebA dataset as an example, we plot in Figure 3 the rank correlation between the rank obtained from reduced sample size and that obtained by 30%[1] of the training set. We can clearly see that even using 5% training samples can accurately estimate the rank, achieving a Spearman rank correlation coefficient of 97.35.

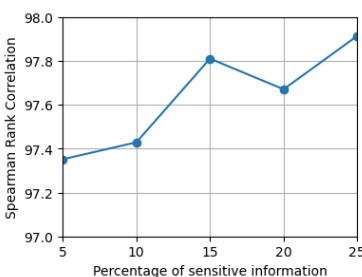

Figure 3: Spearman rank correlation with different percentages of sensitive information.

The MMD scores, $m(z_i, a|\mathbf{u})$ and $m(z_i, y|\mathbf{u})$, reflect the correlation of the $i$-th dimension in $\mathbf{z}|\mathbf{u}$ with the sensitive information $a$, and the target label $y$, respectively. From a fairness perspective, we are looking for the representations that are independent with $a$, but dependent on $y$.

We iterate over all the dimensions in $\mathbf{z}|\mathbf{u}$, calculate MMD scores, and sort them accordingly. We denote $\mathbf{s}_{a,\mathbf{u}} := \arg \mathrm{sort}(-m(\mathbf{z}, a|\mathbf{u}))$, where $\mathbf{s}_{a,\mathbf{u}} \in \mathbb{R}^n$ represents the indices of the sorted MMD scores, arranged in descending order based on sensitive information. Similarly, we denote $\mathbf{s}_{y,\mathbf{u}} := \arg \mathrm{sort}(-m(\mathbf{z}, y|\mathbf{u}))$, where $\mathbf{s}_{y,\mathbf{u}} \in \mathbb{R}^n$ represents the indices of the sorted MMD scores based on target labels.

We define a threshold $t$ to identify the top $t$ indices in $\mathbf{s}_{y,\mathbf{u}}$, which are denoted as $\mathbf{s}_{y,\mathbf{u}}^t$. Given the disentangled nature of our method, the MMD scores across different dimensions show significant differences (see Figure 5). Therefore, we choose $t$ such that the MMD score of the $t$-th dimension exceeds the average MMD score of all dimensions and average on all condition variable:

---

[1]We use 30% for comparison since we found that the rank of sensitive-related dimensions remains consistent using over 30% of the training set.

$$\mathbf{s}_{y|\mathbf{u}}^t = \arg\max_{\mathbf{k}} \sum_{i \in \mathbf{k}} m(z_i, y|\mathbf{u}) \quad s.t. \quad |\mathbf{k}| = t, \tag{4}$$

$$m(z_{\mathbf{s}_{y,\mathbf{u}}[t_\mathbf{u}+1]}, y|\mathbf{u}) < \mathbb{E}[m(\mathbf{z}, y|\mathbf{u})] \leq m(z_{\mathbf{s}_{y,\mathbf{u}}[t_\mathbf{u}]}, y|\mathbf{u}), \quad t = \mathbb{E}[t_\mathbf{u}]$$

In consideration of fairness, we remove the dimensions related to sensitive information from the representation. We denote a hyperparameter $\lambda$, where $\lambda \in \mathbb{R}^+$ to regulate the amount of sensitive information retained in the representation. A larger value of $\lambda$ results in the removal of a greater number of dimensions related to sensitive information.

$$\mathbf{s}_{a|\mathbf{u}}^{\lambda t} = \arg\max_{\mathbf{k}} \sum_{i \in \mathbf{k}} m(z_i, a|\mathbf{u}) \quad s.t. \quad |\mathbf{k}| = \lambda t \tag{5}$$

**Downstream tasks.** The learned representation is applicable to different downstream tasks, we take fair classification as an example. Disentanglement restricts the mutual information between different dimensions in $\mathbf{z}$. Hence, we build a fair classifier $g(\cdot)$ that relies on target-related information while minimizing the influence of sensitive information.

$$\mathbf{s} = \sum_{\mathbf{u}} (\mathbf{s}_{y|\mathbf{u}}^t \setminus \mathbf{s}_{a|\mathbf{u}}^{\lambda t}), \quad \hat{y} = g(\mathbf{z_s}), \tag{6}$$

The objective for a fair classifier is:

$$L(\theta_c) = \arg\min_{\theta_c} \frac{1}{N} \sum_{t=1}^{N} \mathcal{L}_{CE}(\hat{y}_t, y_t), \tag{7}$$

where $N$ is the number of samples, $\theta_c$ is the parameter of the classifier, and $y_t$ is the target label.

**Remark (Fulfills the requirement of fair representation learning):** Fair representation learning requires (1) maximal preservation of data information and (2) statistical independence from sensitive attributes. For (1), our approach is inspired by a theoretical breakthrough in nonlinear ICA, which proves the possibility of reconstructing latent variables through a generative process (Khemakhem et al., 2020), thus ensuring the preservation of data information. For (2), we utilize MMD, a widely used metric for measuring statistical independence (Deka & Sutherland, 2023), to remove dimensions correlated to sensitive attributes, satisfying representation independence of sensitive attributes under MMD metric. We summarize our training step in Algorithms 1 and 2 in the Appendix.

## 4 EXPERIMENT

### 4.1 EXPERIMENT SETUP

We validate our approach on three datasets: UCI Adult (Becker & Kohavi, 1996) CelebA (Liu et al., 2015), and UTKFace (Zhang & Qi, 2017). Each dataset is divided into Group-labeled and Group-unlabeled sets, in line with the limited demographics fairness research (Nam et al., 2022; Jung et al., 2022; Zhang et al., 2022). We provide an introduction to datasets in the Appendix B.

The comparison methods in our study are categorized based on their usage of sensitive information. The first category, which does not utilize sensitive information, includes Empirical risk minimization (**ERM**), **DRO** (Hashimoto et al., 2018), **ARL** (Lahoti et al., 2020); The second category, employing limited sensitive information as **Ours**, comprises **BiFair** (Ozdayi et al., 2021), **JTT** (Liu et al., 2021), **SSA** (Nam et al., 2022), **CGL** (Jung et al., 2022), **NIFR** (Kehrenberg et al., 2020), **FairLISA** (Liu et al., 2024). Please refer to Appendix C and E for detailed implementation.

**Metrics.** We report results through two metrics: Accuracy and Fairness. For the fairness notion, we apply demographic parity (DP) (Dwork et al., 2012), equal opportunity (EOp) (Hardt et al., 2016), and equalized odds (EOd) (Hardt et al., 2016) to measure the degree of fairness.

**Implementation.:** For image datasets, we use the GLOW model as an encoder to learn the latent representation. The GLOW model consists of 4 blocks for both CelebA and UTKFace datasets. Each block contains 32 iterations of flow steps. We follow the origin paper of GLOW and implement one flow step consisting of Actnorm, $1 \times 1$ convolution, and affine coupling. We resize the image to $64 \times 64$. The parameter of GLOW is optimized by Adam optimizer (Kingma & Ba, 2014), with a learning rate $10^{-4}$. We show the training details in section E of the Appendix.

## 4.2 RESULTS

### 4.2.1 TABULAR RESULTS

We first validate our method's flexibility to protect different sensitive groups within a single representation with the tabular dataset. We report results on protecting $a$ = gender in the Adult dataset in Table 1a. Among the evaluated baselines, our method exhibited the highest performance on DP and EOd, with a slight accuracy loss. It indicates that our method is capable of effectively disentangling the sensitive information $a$ from the target label $y$, and enhancing fairness in tabular dataset.

We apply the same representation[2] to the task focused on protecting $a$ = race, results are in Table 1b. Our method yielded similar outcomes and achieved better fairness compared to methods that require retraining for different protected attributes. This demonstrates its capability to effectively protect various sensitive attributes using a single representation.

Table 1: Classification results of Adult dataset

| | DP↓ | EOp↓ | EOd↓ | ACC↑ | | DP↓ | EOp↓ | EOd↓ | ACC↑ |
|---|---|---|---|---|---|---|---|---|---|
| ERM | 17.70±0.52 | 9.78±1.98 | 8.56±1.14 | **84.54±0.05** | ERM | 8.84±0.39 | 4.29±0.30 | 3.83±0.34 | 84.47±0.10 |
| DRO | 17.85±0.25 | 9.54±0.30 | 8.54±0.14 | 84.38±0.02 | DRO | 9.11±0.19 | 4.45±0.31 | 4.04±0.22 | 84.38±0.03 |
| ARL | 17.58±0.04 | 10.12±1.03 | 8.68±0.45 | 84.51±0.04 | ARL | 8.66±0.08 | 4.48±0.14 | 3.82±0.10 | **84.56±0.03** |
| JTT | 15.70±1.28 | **1.67±0.80** | 4.08±0.17 | 83.90±0.24 | JTT | 8.82±0.50 | 2.95±0.84 | 3.24±0.36 | 83.88±0.25 |
| BiFair | 17.04±1.71 | 7.24±2.95 | 7.14±2.06 | 84.19±0.05 | BiFair | 8.25±0.89 | 4.57±0.79 | 3.73±0.70 | 84.24±0.17 |
| SSA | 16.43±0.99 | 3.99±2.32 | 5.40±0.83 | 80.71±0.52 | SSA | 11.93±0.24 | 1.12±0.89 | 3.89±0.49 | 81.68±0.18 |
| CGL | 16.22±0.29 | 4.35±0.64 | 5.44±0.27 | 81.27±0.44 | CGL | 12.37±0.28 | **0.34±0.29** | 3.84±0.03 | 80.69±0.36 |
| NIFR | 10.61±0.50 | 4.82±3.06 | 3.34±1.97 | 82.52±0.13 | NIFR | 8.19±0.41 | 3.39±0.53 | 2.97±0.64 | 84.01±0.19 |
| LISA | 10.78±0.48 | 3.34±0.27 | 2.46±0.18 | 82.83±0.22 | LISA | 7.40±0.37 | 3.44±0.64 | 3.02±0.18 | 83.51±0.50 |
| Ours | **10.53±0.19** | 3.27±0.17 | **1.82±0.04** | 82.59±0.24 | Ours | **6.23±0.15** | 2.27±0.47 | **1.89±0.28** | 83.03±0.02 |

(a) $a$ = gender, $y$ = income.  (b) $a$ = race, $y$ = income.

### 4.2.2 VISION RESULTS

In Tables 2a and 2b, we present results for the CelebA dataset. A key distinction of our approach is the use of the same representation for both tasks, unlike most other methods which, apart from NIFR, need to be retrained for each task. Notably, our method demonstrates an improvement in fairness metrics in Table 2a and maintains utility compared to other methods that require an equivalent amount of sensitive information. For the second task, while CGL outperforms EOp, it exhibits variability in performance across different runs. The effectiveness of SSA in balancing fairness and utility depends on the accuracy of pseudo-sensitive label predictions. SSA performs well when the accuracy of predicting sensitive information is high. However, the performance of SSA is unstable, showing sub-optimal fairness results as shown in Table 2a.

Table 2: Classification results on CelebA

| | DP↓ | EOp↓ | EOd↓ | ACC↑ | | DP↓ | EOp↓ | EOd↓ | ACC↑ |
|---|---|---|---|---|---|---|---|---|---|
| ERM | 15.23±0.72 | 11.52±3.05 | 9.76±1.50 | **82.66±0.24** | ERM | 16.81±0.44 | 46.61±4.17 | 24.52±1.96 | **95.05±0.06** |
| DRO | 14.45±2.66 | 12.18±0.62 | 10.24±0.87 | 82.17±0.29 | DRO | 18.89±0.70 | 40.71±6.09 | 22.40±2.73 | 92.99±0.99 |
| ARL | 9.28±2.21 | 10.62±0.79 | 8.08±0.97 | 80.68±0.20 | ARL | 18.29±3.39 | 38.92±1.23 | 21.83±1.53 | 91.64±1.33 |
| JTT | 13.47±0.59 | 6.66±0.76 | 7.83±0.39 | 78.44±0.76 | JTT | 17.76±0.32 | 37.54±1.07 | 19.89±0.32 | 92.78±0.17 |
| BiFair | 4.07±1.17 | 5.34±1.57 | 3.89±1.14 | 79.67±0.17 | BiFair | 10.31±2.01 | 32.68±4.16 | 16.71±2.21 | 90.81±0.77 |
| SSA | 13.19±0.66 | 5.41±0.64 | 6.82±0.19 | 80.64±0.17 | SSA | 17.07±0.51 | 20.63±1.18 | 11.15±0.87 | 92.85±0.83 |
| CGL | 14.39±0.78 | 2.02±0.14 | 5.26±0.36 | 80.67±0.45 | CGL | 14.70±4.52 | **16.14±6.21** | 10.01±2.19 | 90.83±1.93 |
| NIFR | 9.43±0.86 | 6.36±0.56 | 5.64±0.15 | 80.79±0.35 | NIFR | 14.98±1.32 | 39.19±0.9 | 21.21±0.53 | 92.28±0.06 |
| LISA | 5.63±0.63 | 2.71±0.18 | 1.95±0.27 | 81.54±0.22 | LISA | 6.95±1.64 | 21.38±5.70 | 11.15±3.14 | 89.74±0.54 |
| Ours | **3.25±0.42** | **1.59±0.17** | **0.99±0.02** | 80.89±0.08 | Ours | **5.18±0.35** | 19.32±0.12 | **9.94±0.07** | 88.98±0.05 |

(a) $a$ = Young, $y$ = Bags under eyes.  (b) $a$ = Male, $y$ = Blond Hair.

In Table 3, we present the evaluation results on the UTKFace dataset. Our method outperforms other baseline models in terms of fairness metrics. Furthermore, it attains a utility performance that is comparable with ERM training. This illustrates not only the effectiveness of our approach in enhancing fairness but also its ability to maintain a high level of utility.

**Difference from disentangled method.** NIFR (Kehrenberg et al., 2020) utilizes disentanglement and incorporates adversarial training to ensure that the debiased latent representation remains unpredictable with respect to sensitive labels. The primary motivation for using adversarial training is to

---

[2]The representation learned through Section 3.2

address the lack of identifiability within the disentangled latent space, which leads to the coupling of sensitive and target information. However, with the identifiable representation learned through nonlinear ICA, we effectively address these coupling issues without adversarial learning, enhancing fairness outcomes by a large margin with utility performance comparable to adversarial approaches.

Table 3: Classification results on UTKFace dataset.

| Method | DP↓ | EOp↓ | EOd↓ | ACC↑ |
|---|---|---|---|---|
| ERM | $19.82_{\pm0.30}$ | $20.37_{\pm2.58}$ | $17.72_{\pm0.25}$ | $82.18_{\pm1.00}$ |
| DRO | $20.48_{\pm1.75}$ | $19.26_{\pm1.59}$ | $18.80_{\pm1.74}$ | $78.09_{\pm0.15}$ |
| ARL | $19.14_{\pm2.78}$ | $19.59_{\pm5.13}$ | $17.52_{\pm2.93}$ | $78.77_{\pm0.38}$ |
| JTT | $17.86_{\pm0.81}$ | $13.20_{\pm1.16}$ | $15.75_{\pm0.82}$ | $82.33_{\pm0.50}$ |
| BiFair | $14.07_{\pm1.02}$ | $11.14_{\pm1.91}$ | $12.07_{\pm1.03}$ | $82.31_{\pm0.43}$ |
| SSA | $13.06_{\pm1.31}$ | $11.14_{\pm1.79}$ | $11.05_{\pm1.15}$ | $\mathbf{83.49_{\pm0.49}}$ |
| CGL | $14.23_{\pm0.94}$ | $9.73_{\pm1.45}$ | $12.07_{\pm0.95}$ | $83.03_{\pm1.29}$ |
| NIFR | $18.43_{\pm0.48}$ | $19.26_{\pm1.00}$ | $16.50_{\pm0.51}$ | $78.88_{\pm0.48}$ |
| FairLISA | $9.58_{\pm0.57}$ | $9.72_{\pm1.42}$ | $7.50_{\pm0.56}$ | $81.99_{\pm1.29}$ |
| Ours | $\mathbf{8.77_{\pm0.05}}$ | $\mathbf{7.43_{\pm0.30}}$ | $\mathbf{6.78_{\pm0.04}}$ | $83.17_{\pm0.04}$ |

**Fairness accuracy trade-off** We explore the impact of the hyperparameter $\lambda$, as defined in Section 3.2. The effects on both accuracy and fairness metrics are illustrated in Figure 4. When $\lambda = 0$, it indicates that the latent representation is disentangled but no sensitive information is removed. This setting improves the fairness metric compared to ERM training, which supports the finding that disentanglement itself can benefit fairness (Locatello et al., 2019a). However, there remains a significant margin compared with methods that remove sensitive information from the latent space, which can further enhance fairness in predictions.

From Figure 4 we observe that with the $\lambda$ increasing, there is a consistent improvement in fairness metrics, accompanied by a small drop in accuracy. In other words, reducing sensitive-related information benefits a fair outcome. Even at $\lambda = 2$, indicating that the sensitive-related dimension is twice the target-related dimension, we can still preserve an accuracy of 88% for one task in CelebA and 82% accuracy for UTKFace datasets. This result indicates that the coupling of $y$ and $a$ is mild, which is significant given that $y$ and $a$ are not used in the representation learning process.

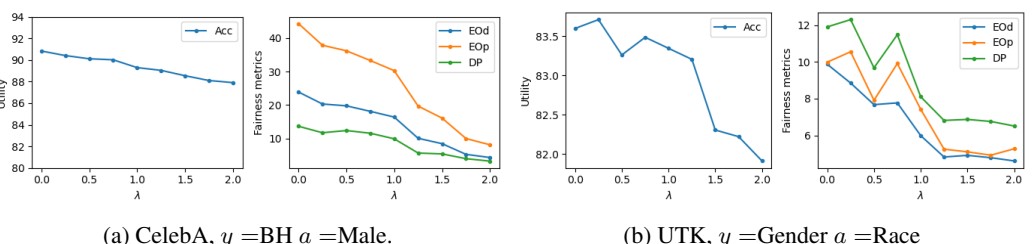

(a) CelebA, $y$ =BH $a$ =Male.  (b) UTK, $y$ =Gender $a$ =Race.

Figure 4: Fairness-accuracy trade-off. BH stands for Blond hair

**Discussion** The objective of the proposed method is to learn a debiased representation that encodes minimal sensitive information, rather than optimizing the representations with respect to a specific fairness metric. From Table 1-3, it is evident that the proposed method achieves optimal results in terms of DP and EOd. However, EOp may not always reach optimal levels. We have conducted experiments evaluating the MMD scores under different fairness notions to further investigate this aspect. For DP, we compute the MMD score using $p(\mathbf{z}|a = 0)$ and $p(\mathbf{z}|a = 1)$. For EOp, we compute the MMD score using $p(\mathbf{z}|y = 1, a = 0)$ and $p(\mathbf{z}|y = 1, a = 1)$. For EOd, we compute MMD in the two scenarios, one is $p(\mathbf{z}|y = 1, a = 0)$ and $p(\mathbf{z}|y = 1, a = 1)$, the other is $p(\mathbf{z}|y = 0, a = 0)$ and $p(\mathbf{z}|y = 0, a = 1)$, then we take an average. The number of samples is the same when computing the MMD score under different fairness considerations. We plot the top 10 MMD scores under different notions in CelebA and UTK datasets, as shown in Figure 5.

Although the MMD score is computed under different fairness considerations, we observe that the dimensions identified share similarities across different fairness notions. *This suggests that by removing the shared dimensions containing sensitive information, we can simultaneously improve all three fairness metrics.* However, EOp shows less overlap with the other two notions, as indicated by the right three orange bars in Figure 5a that do not align with the other two bars. This discrepancy leads to less comparable results in EOp. Nonetheless, it is important to note that modifying a single fairness metric is not the primary objective of this work.

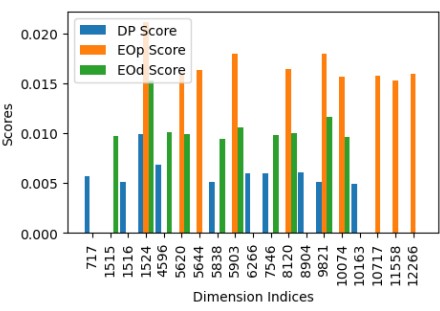 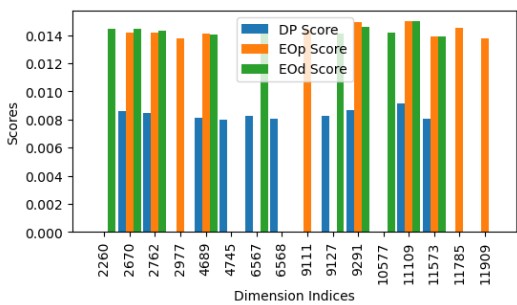

(a) CelebA dataset  (b) UTK dataset

Figure 5: Top 10 MMD scores indices computed under various fairness considerations. The dimensions related to different fairness notions are interwoven (e.g., Dimension 1524 in CelebA is associated with all considered fairness notions).

### 4.2.3 DISENTANGLED IMAGE GENERATION

In addition to fair classification tasks, Our approach can also be employed in the field of image generation. We visualize the generated images by manipulating the latent dimension in $\mathbf{z}$ with the highest correlation to a certain attribute. Concretely, we screen the dimension in the latent space with the highest MMD score. We randomly feed samples into the INN model to extract the latent representation. We alter the value of the identified most sensitive-related row and feed the altered latent representation back to the inverse of the INN model to generate an image, as shown in Figure 6. Our results indicate that a change in a single row of the latent space is sufficient to alter certain attributes of the synthesized image. The difference between our method and the GLOW for manipulating images with an attribute lies in the update direction. GLOW computes the direction using $\mathbf{z}_{pos} - \mathbf{z}_{neg}$, where $\mathbf{z}_{pos} = \mathbb{E}[\mathbf{z}|att = 1]$ and $\mathbf{z}_{neg} = \mathbb{E}[\mathbf{z}|att = 0]$, $att$ is the attribute targeted to change. This approach potentially involves all dimensions in $\mathbf{z}$. In contrast, our method, which is theoretically identifiable, manipulates only one dimension in the latent space to demonstrate semantically meaningful changes.

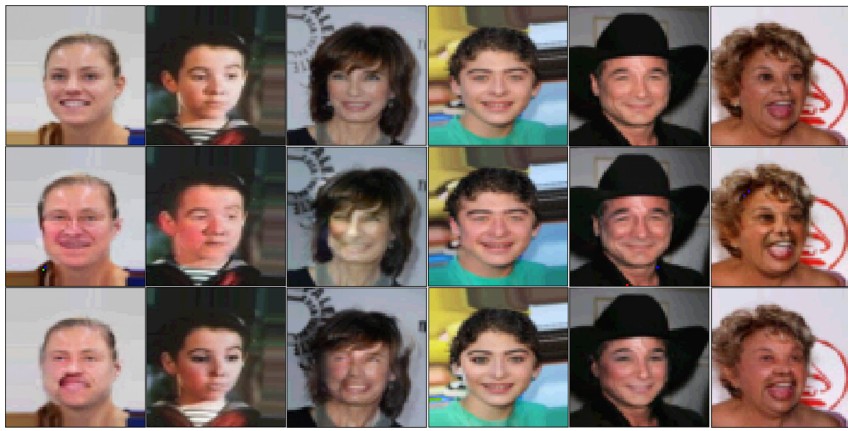

Figure 6: Disentanglement effects in CelebA dataset: The first row shows the original images, the second row shows changes in *age*, and the third row illustrates changes in *gender*. All synthetic images are generated by manipulating only one dimension in the latent space.

## 5 CONCLUSION

In this paper, we utilize the nonlinear ICA to find a theoretically guaranteed disentangled representation and build a fair classifier based on the representation. The proposed method is sensitive information free for learning disentangled latent representation, and requires a small portion of the sensitive information to capture the sensitive related dimension in the latent space. Even if the sensitive information changes, we can adapt the representation without retraining. We evaluate our method on three real-world datasets and compare it with the state-of-the-art methods. We empirically show that our approach promotes group fairness in different tasks. Future research could be the generalization of the method, like exploring its application in fair clustering tasks.

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

## A  ALGORITHM

We summarize our training algorithm in the Algorithm 1 and 2 .

---

**Algorithm 1** Learning disentangled representations.

---

**Input:** Input data $\mathbf{x} \in \mathcal{X}$, auxiliary variable $\mathbf{u}$.
**Output:** Encoder $f_\theta$.
 1: **for** $i = 0 : T - 1$ **do**
 2:     $\mathbf{z} = f_\theta^{-1}(\mathbf{x}, \mathbf{u})$, where $f_\theta$ is a flow-based model
 3:     Update $\theta$ by minimizing objective function in equation 2
 4: **end for**

---

**Algorithm 2** Identifying latent representations.

---

**Input:** $v\%$ sampled data with sensitive information $(\mathbf{x}, a) \in \mathcal{X}_s \times \mathcal{A}_s$, where $|\mathcal{X}_s| = \frac{v}{100}|\mathcal{X}|$. A hyperparameter $\lambda$, auxiliary variable $\mathbf{u}$.
**Output:** Fair latent representation $\mathbf{z}_{out}$.
 1: **for** $\mathbf{x}$ in $\mathcal{X}_s$ **do**
 2:     $\mathbf{z}_{\text{sample}} = f_\theta^{-1}(\mathbf{x})$, where $\mathbf{z}_{\text{sample}} \in \mathbb{R}^n$, $f_\theta$ is learned by Algorithm 1.
 3: **end for**
 4: Split latent representation into four sets $Z_{a0} = \{\mathbf{z}_{\text{sample}}|a = 0\}$, $Z_{a1} = \{\mathbf{z}_{\text{sample}}|a = 1\}$, $Z_{y0} = \{\mathbf{z}_{\text{sample}}|y = 0\}$, and $Z_{y1} = \{\mathbf{z}_{\text{sample}}|y = 1\}$.
 5: **for** $u = 1$ to $|\mathbf{u}|$ **do**
 6:     **for** $i = 1$ to $n$ **do**
 7:         $m(z_i, a|u) = \text{MMD}(\mathcal{F}, p(z_i|a = 0, u), p(z_i|a = 1, u))$.          $\triangleright$ Compute MMD score based on $Z_{a0}$ and $Z_{a1}$.
 8:         $m(z_i, y) = \text{MMD}(\mathcal{F}, p(z_i|y = 0, u), p(z_i|y = 1, u))$.          $\triangleright$ Compute MMD score based on $Z_{y0}$ and $Z_{y1}$.
 9:     **end for**
10: **end for**
11: $\mathbf{s}_{a|\mathbf{u}} = \arg \text{sort}(-m(z_i, a|\mathbf{u}))$, $\mathbf{s}_{y|\mathbf{u}} = \arg \text{sort}(-m(z_i, y|\mathbf{u}))$
12: $\mathbf{s}_{y|\mathbf{u}}^t = \arg \max_\mathbf{k} \sum_{i \in \mathbf{k}} m(z_i, y|\mathbf{u})$,   s.t. $|\mathbf{k}| = t$.
13: $\mathbf{s}_{a|\mathbf{u}}^{\lambda t} = \arg \max_\mathbf{k} \sum_{i \in \mathbf{k}} m(z_i, a|\mathbf{u})$, s.t. $|\mathbf{k}| = \lambda t$.
14: $\mathbf{s} = \sum_\mathbf{u} \mathbf{s}_{y|\mathbf{u}}^t \setminus \mathbf{s}_{a|\mathbf{u}}^{\lambda t}$
15: **for** $\mathbf{x}$ in $\mathcal{X}$ **do**
16:     $\mathbf{z} = f_\theta^{-1}(\mathbf{x})$
17:     $\mathbf{z}_{out} = \mathbf{z_s}$
18: **end for**

---

## B  DATASET INTRODUCTION

The CelebA (Liu et al., 2015) dataset is a large-scale facial dataset featuring 40 annotated attributes. In our study, we focus on well-established settings in fairness research (Wang et al., 2022; Morales et al., 2020), including $y =$ Blond hair, $a =$ male. $y =$ Bags under male, $a =$ Young. The data distributions in the training set are shown in Figure 7.

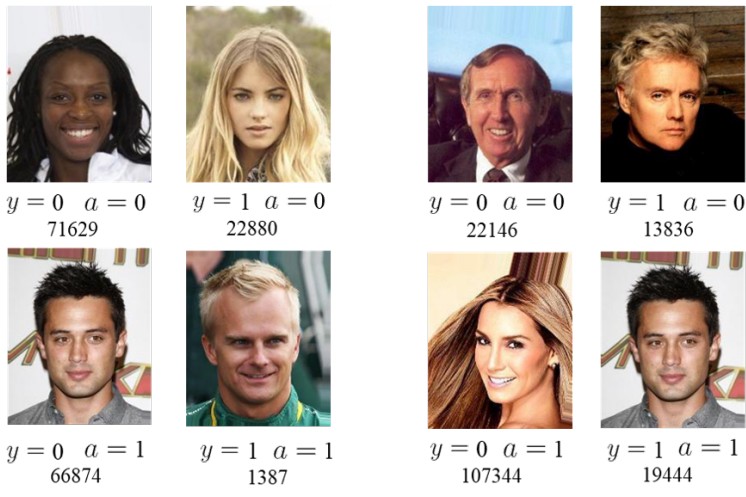

Figure 7: The training sample distribution in the CelebA dataset.

For the CelebA dataset, the Group-unlabeled dataset comprises 159515 samples, and the Group-labeled dataset contains 3255 samples. There is no overlap between these two datasets. Additionally, the test set is divided according to the default split.

The UTKFace (Zhang & Qi, 2017) dataset is a large-scale face dataset, annotated with three attributes: age (ranging from 0 to 116 years), gender (male and female), and ethnicity (White, Black, Asian, Indian, and Others). In our study, we focus on the task with $y$ =gender, $a$ = race, where we binarize the race attribute into {White, non-White} for sensitive labels. The distribution of training samples is shown in Figure 8. The Group-unlabeled dataset has 6101 samples, and the Group-labeled dataset has 3556 samples. The test set has 3555 samples.

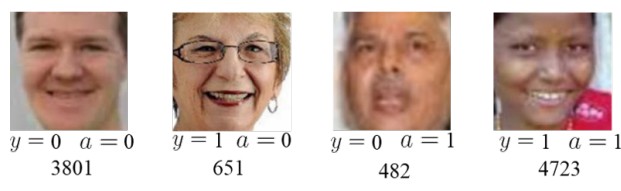

Figure 8: The training sample distribution in the UTKFace dataset

UCI Adult (Becker & Kohavi, 1996) is a tabular dataset. The primary task is to predict whether an individual earns more than $50,000$ per year. It includes 14 attributes that describe various aspects of each individual, such as age, work class, education, marital status, occupation, relationship, race, sex, capital gain, capital loss, hours per week, native country, and income classification. The distribution of training samples is shown in Figure 9. For the Adult dataset, the Group-unlabeled dataset includes 27145 samples, the Group-labeled dataset has 3017 samples, and the test set consists of 15060 samples.

## C  BASELINE METHODS IMPLEMENTATION

In this section, we detail the model architectures and hyperparameters used by each baseline approach.

|       | $y$=0 | $y$=1 |
|-------|-------|-------|
| $a$=0 | 19094 | 6839  |
| $a$=1 | 3560  | 669   |

(1) $y$ = income, $a$ = race

|       | $y$=0 | $y$=1 |
|-------|-------|-------|
| $a$=0 | 8670  | 1112  |
| $a$=1 | 13984 | 6396  |

(2) $y$ = income, $a$ = sex

Figure 9: The training sample distribution in the Adult dataset

## C.1 IMAGE DATASETS

For a fair comparison, we employ ResNet-101 (He et al., 2016), whose complexity is comparable to that of GLOW, as the backbone model for both the CelebA and UTKFace datasets. We do not incorporate pre-trained weights in our implementation. We resize each image to $64 \times 64$, and augmentation techniques are not employed. For the baseline implementation, we turn all hyperparameters based on the optimal accuracy and fairness trade-off as introduced in section 4.1 on the group-labeled dataset.

**ERM**: For both CelebA and UTKFace datasets, the ERM model is trained by utilizing a cross-entropy loss function. We use Adam optimizer to update the parameters of the network, with a learning rate of $1 \times 10^{-4}$. We train the ERM model for 30 epochs with batch size 128.

**DRO**: In the implementation of Distributionally Robust Optimization (DRO), we employ the identical backbone model as utilized in the ERM model. We turn the hyperparameter $\eta \in \{0.1, 0.2, 0.3\}$ to control the threshold of the loss for optimization. We select $\eta = 0.2$ as the optimal setting for both the CelebA and UTKFace datasets.

**ARL**: For the Adversarial Representation Learning (ARL) approach, an auxiliary network is necessary. Following the original author's suggestion, we implement this using a smaller-scale network, specifically selecting ResNet-18 as the adversarial network without loading pre-trained weights. The auxiliary network is optimized using Adam optimizer with a learning rate of $10^{-5}$.

**JTT** For the Just train twice (JTT) method, the backbone model is the same as the ERM model. For the CelebA dataset, we follow the original paper's implementation, setting the hyperparameter $\lambda_{up} = 50$ and $T = 1$. For the UTKFace dataset, we adjust these hyperparameters to $\lambda_{up} = 20$ and $T = 10$. The initial training phase utilizes the Adam optimizer with a learning rate of $10^{-4}$. For the second training phase, we use the Adam optimizer with a learning rate of $10^{-4}$ and a weight decay of $10^{-4}$ as suggested in the paper (Liu et al., 2021).

**BiFair**: In implementing the BiFair method, we maintain the same model architecture as utilized in the baseline models. For the CelebA and UTKFace datasets, we set the hyperparameters with $T_{in} = 80$, $T_{out} = 20$, and $\lambda = 0.8$. We use the Adam optimizer unchanged as the official code (Ozdayi et al., 2021). The inner optimizer is configured with a learning rate of $10^{-3}$, and the learning rate for the outer optimizer is set at $10^{-4}$.

**SSA**: In the implementation of the SSA, we utilize the ResNet-101 architecture for both the pseudo-sensitive label assignment and the robust optimization phases. During the pseudo-sensitive labeling phase, the Adam optimizer is employed with a learning rate of $10^{-3}$, and we train 50 epochs. We set $\tau_{g_{min}} = 0.95$ as the same as the official implementation (Nam et al., 2022); For the robust training phase, we adopt the Group-DRO method for robust optimization, following the authors' recommendation. We use Adam optimizer with a learning rate of $10^{-4}$, $l_2$ regularization $10^{-4}$, and training for 50 epoch.

**CGL**: The CGL method encompasses two training phases. we employ ResNet 101 for both stages as recommended in the original paper (Jung et al., 2022). We utilize the officially released code to implement the CelebA and UTKFace datasets. We use the Adam optimizer with a learning rate of $10^{-4}$. In the second phase, for consistency in comparison, we adopt Group-DRO as the robust training technique, aligning with the approach used in the SSA.

**NIFR**: The NIFR, a flow-based model like ours, is implemented using the GLOW (Kingma & Dhariwal, 2018) model as recommended in the paper (Kehrenberg et al., 2020). For latent encoding, we adhere to the official code, selecting 12 dimensions in $\mathbf{z}$ to represent sensitive information and

12276 dimensions for non-sensitive data. In the training phase, we choose $a = $ [male, Young] to develop a representation that ensures fairness with respect to these sensitive attributes in the CelebA dataset. During the classifier training stage for the CelebA dataset, the same $\mathbf{z}$ is utilized to train distinct classifiers for two separate tasks. The optimizer is used Adam, with a learning rate $10^{-4}$ for training the GLOW model. For UTKFace dataset, we choose $a = $ race, and the classifier structure is the same as the classifier used in CelebA dataset.

**FairLISA**: In implementing FairLISA, we utilize a ResNet 101 architecture for the Filter, aligning with the baseline models, and a 2-layer MLP with hidden nodes [16, 16] for the discriminator. We adhere to the official code's hyperparameters, setting $\lambda_1 = 1$, $\lambda_2 = 1$, and $\lambda_3 = 0.5$. During each iteration, we train the discriminator for 10 steps ($T = 10$). Both filter and discriminator are optimized using the Adam optimizer, with a learning rate of $10^{-4}$ for each.

## C.2 TABULAR DATASET

For the tabular dataset, we utilize a simple Multi-Layer Perceptron (MLP) as the backbone model for all methods. This MLP comprises two layers with 32 and 64 nodes respectively, and employs ReLU activation. For ERM, the Adam optimizer is used with a learning rate of $10^{-3}$ and batch size of 256.

For DRO, we set the hyperparameter $\lambda = 0.1$. For ARL, the auxiliary network is implemented as a linear layer, in accordance with the original paper (Lahoti et al., 2020).

For JTT, we tune the hyperparameters to $T = 1$ and $\lambda_{up} = 2$, other hyperparameters remain consistent with the ERM model. For BiFair, we select hyperparameters as $T_{out} = 40$, $T_{in} = 20$, and $\lambda = 0.8$.

For both SSA and CGL, during the pseudo-sensitive labeling step, we use the Adam optimizer with a learning rate of $10^{-3}$ for 20 epochs. For the second stage, Group DRO is employed as the optimization objective, with the Adam optimizer at a learning rate of $10^{-3}$, and a training duration of 50 epochs.

For NIFR, we allocate 6 dimensions for sensitive information encoding and 26 for non-sensitive information encoding. As NIFR is also implemented on the Adult dataset, the structure of the learning network remains unchanged. It contains 1 level and 35 coupling channels (Kehrenberg et al., 2020). For classifier training, we employ a 2-layer MLP consistent with the baseline models. The classifier is optimized using the Adam optimizer with a learning rate of $10^{-3}$.

For FairLisa, we employ the same classifier as used in ERM training to serve as the filter. The discriminator is constructed using a one-layer neural network with 16 nodes. We set the hyperparameter $\lambda$ consistently with the image dataset settings. To ensure stable training, we utilize the Adam optimizer with a learning rate of $10^{-4}$ to optimize both networks.

## D EXPANDED DISCUSSION ON AUXILIARY VARIABLE SELECTION

The objective of disentangled representation learning is to develop a broadly applicable representation for downstream tasks. We expect a single representation should be adaptable to meet the requirements of various tasks while protecting different sensitive attributes. Therefore, the selection of $\mathbf{u}$ is critical. As discussed in Section 3.2, we extend our justification to the task of protecting the attribute 'Young'. The Pearson correlation coefficients and balance ratio for this are shown in Figure 10. We shadowed the attribute that served as the auxiliary variable. It is clearly observable that the selected attribute still fulfills the criteria. Therefore, this guarantees that the learned representation is capable of disentangling different sensitive information across the latent dimensions. The disentanglement effect is demonstrated in Section 4.2.3, where two distinct dimensions capture gender and age information.

## E IMPLEMENTATION DETAILS

All experiments were conducted on a single Nvidia RTX 3090 GPU with 24 GB of memory.

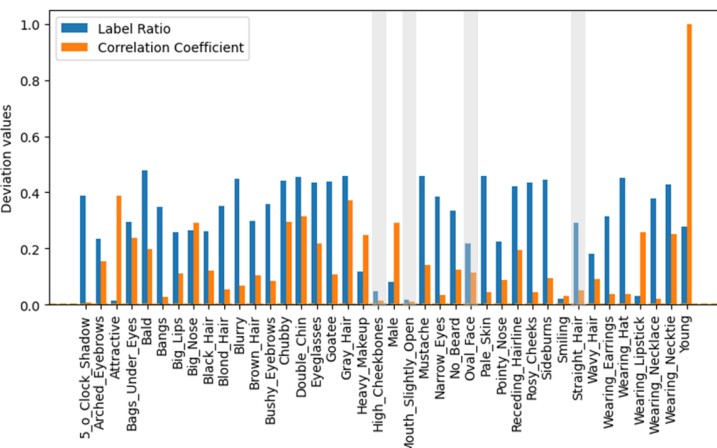

Figure 10: Bar plot of Pearson correlation coefficients and balance ratios in the CelebA dataset. Orange bars denote the absolute value of Pearson correlation coefficients between annotations and 'Young', with bars nearing 0 indicating less correlated variables. Blue bars illustrate the balance ratio for each annotation, with values close to 0 indicating higher balance.

**Image Dataset**  For the CelebA and UTKFace datasets, we utilize GLOW to implement the invertible neural network that learns the representations introduced in Section 3.2. The foundational architecture of the GLOW model consists of a sequence of flow layers. Each flow layer consists of an Actnorm layer, a $1 \times 1$ convolution layer, and an affine coupling. For the CelebA and UTKFace datasets, we select the hyperparameters for the depth of the flow, $K = 32$, and the number of levels, $L = 4$.

For the CelebA dataset, we select the auxiliary variable $\mathbf{u} = \{$*High Cheekbones*, *Mouth Slightly Open*, *Straight Hair*, *Oval Face*$\}$. Each sample is associated with a corresponding auxiliary label, yielding a total of $2^4 = 16$ possible combinations of auxiliary variables. It should be noted that a large number of auxiliary variables is generally necessary to achieve identifiability, according to Khemakhem et al. (2020). However, Sorrenson et al. (2020) observed that the identifiability property can still be maintained with a smaller number of auxiliary variables, indicating that a large number is not a necessary condition. For the hyperparameter $\lambda$, we consider the set $\lambda \in [0.25, 0.5, 1, 1.25, 1.5]$. The optimal value of $\lambda$ is selected to maximize the objective function defined as $Acc \times (1 - EOd)$ within the group-labeled dataset. Specifically, for the CelebA dataset, the optimal $\lambda$ is determined to be 1.5.

For UTKFace dataset, the auxiliary variable $\mathbf{u}$ is determined based on age. Specifically, we categorize the age labels into four groups: the first group contains ages from $[0, 20)$, the second group from $[20, 40)$, the third from $[40, 60)$, and the fourth includes ages $[60, 116]$. Within each batch, samples are grouped based on $\mathbf{u}$, and we compute statistics of each group in the latent space, minimizing the objective function as defined in Eq 2. The network parameters are updated using the Adam optimizer with a learning rate of $1 \times 10^{-4}$, and the model is trained for 200 epochs. The hyperparameter $\lambda = 1$ in the UTKFace dataset.

For the image dataset in the fair classification task, we employ a single-layer perceptron, which proves sufficient for performing the classification. This classifier is optimized using an SGD optimizer with a learning rate of 0.09. In task transitions, such as within the CelebA dataset where the objective shifts to predicting 'blonde hair' while protecting gender information, we apply the method outlined in Section 3.2. This requires only retraining the single-layer perceptron.

**Tablar dataset**  For the tabular dataset, we use RealNVP(Dinh et al., 2016) to learn the latent representation (Sorrenson et al., 2020; Dinh et al., 2016). The backbone of our model comprises 18 coupling blocks, with each block being constructed using a 2-layer MLP. Both layers in these blocks consist of 32 nodes and utilize ReLU nonlinearity activations.

For the Adult dataset, we select the auxiliary variable $\mathbf{u} = $ Work class $\in \{$ *Private*, *Self-emp*, *Gov*, *Without-pay* $\}$. We utilize the Adam optimizer with a learning rate of $10^{-3}$ and a batch size of 256, and training for 200 epochs. The hyperparameter $\lambda = 1$ in the Adult dataset.

## F    LIMITATION

This work tackles fairness in representation learning under limited access to sensitive labels. We propose a flow-based model leveraging non-linear ICA for fair representation, mitigating biases while achieving competitive performance on large-scale datasets. However, given the nature of the generative model, our methods need a relatively large-scale dataset to learn a high-quality latent representation.

