# OpenReview forum: "Learning Disentangled Representations for Fairness with Limited Demographics"
_ICLR.cc/2025/Conference — ICLR 2025 Conference Withdrawn Submission_

### Official Review · Reviewer_8pXy · 2024-10-28

**Soundness:** 2
**Presentation:** 4
**Contribution:** 2
**Rating:** 5
**Confidence:** 3

**Summary:**

This paper proposed a fair representation learning method specifically tackle the problem when sensitive information is limited provided by employing the Nonlinear ICA technique to disentangle the sensitive information $a$ from representation $z$ by utilizing the auxilarity variable $u$ (uncorelate to $a$ and balanced...) and further use partial $a$ train MMD score to define which dimension is related to $a$, and remove them (the number of dimensions are defined by $\lambda$.)

**Strengths:**

This paper utilize the technique of Non-linear ICA and MMD to disentangle the sensitive information from the learned representation $z$. The idea of employing an auxiliary variable to achieve disentanglement and further using the MMD score to identify sensitive-correlated dimensions is quite interesting and the overall framework is easy to follow and understand, and the method is convenient for practical applications (the deployment). The experimental results are solid and comprehensive, covering the most popular fairness criteria (demographic parity, equalized odds, and equal opportunity) across different tasks (tabular data and image data, although the results on tabular data are not particularly convincing…) and demonstrate the effectiveness compared to several closely related baseline models.

**Weaknesses:**

- My main concern with the proposed method is around “limited fairness information,” which is the key condition or problem this paper aims to address. The author spent a lot of effort in the introduction discussing how existing works address this issue. However, in this paper, the description of this part is limited a few words (lines 295 to 307) and is highly empirical ('we empirically justify that only a few samples are needed to search the sensitive-related dimensions in the latent space'). Overall, whether it’s the use of nonlinear ICA, the calculation of MMD scores, or the focus on removing sensitive information in representation learning, the author does not emphasize how this method is related to or specifically addresses the problem of **limited sensitive information**. The relationship between these aspects could be made more explicit.

- The captions of each figure and table could be extended with a few words to provide a brief explanation (very minor, just a suggestion).

- As stated in the limitations section, the method requires a relatively large-scale dataset to learn a high-quality latent representation, so the experimental results on the tabular data (which is relatively small compared to the two image datasets) are not as strong as those on the image datasets. It would be interesting to include more explanation, such as convergence analysis, regarding the method, for example, discussing the convergence bound related to sample size.

- “Each dataset is divided into Group-labeled and Group-unlabeled sets, in line with the limited demographics fairness research......” It would be better to move the information from appendix b into the main manuscript to provide an overview of the ratio between group-labeled and group-unlabeled sets.

- Lack of an experiment on how the ratio of group-labeled to group-unlabeled samples affects the results....

**Questions:**

How are the group-labeled and group-unlabeled datasets sampled? Are they randomly split from the original training set, or is a specific proportion of samples randomly selected from each demographic group?

---

### Official Review · Reviewer_gUAP · 2024-11-02

**Soundness:** 2
**Presentation:** 2
**Contribution:** 2
**Rating:** 3
**Confidence:** 4

**Summary:**

The authors propose a two-step framework, which utilizes Non-linear Independent Component Analysis (Nonlinear ICA) to learn disentangled representation. Among the entire process, only a small portion of sensitive information is required in the second step to learn a fair representation.

**Strengths:**

1. The paper gives a comprehensive review on related work, including learning with limited sensitive information and fair representation learning.
2. The authors provide extensive experiments to prove the performance of proposed method, including the tabular/non-tabular data.

**Weaknesses:**

- Limited discussion/theoretical support of "sensitive-information-free": as claimed by the authors, the proposed method only uses a very small fraction of sensitive information, which will make it preferable over other methods. While in the paper, it seems the only evidence (in terms of the fraction of training samples $v$) is from the Figure 3, which is merely from one dataset. A major question is: how is the overall performance linked with the fraction $v$ of sensitive information used to identify latent representations? More discussion (or theoretical insights) should be provided.

- Experiments: the experiments show the effects of $\lambda$, which controls the number of latent variables to be removed. However, no experiment shows the effects of $v$, which is the fraction of samples containing sensitive information. To investigate the performance of the proposed method with limited sensitive information given, the authors may need more empirical evidence showing the proposed method will provide better results than others while controlling $v$. For example: comparing the proposed method to all baselines across a range of $v$ values (e.g. 1%, 5%, 10%, 25%) on multiple datasets.

- Scalability/Efficiency: in the algorithm, it needs computing MMD for each $z_i | a, u$, how does it scale well with multiple $a, u$ (and potentially $y$ when considering equalized odds or equal opportunity)? The authors may need to discuss the scalability (or computational efficiency) and compare it with other benchmarks. For example: time and memory complexity analysis of the algorithm as a function of number of attributes, auxiliary variables, and dataset size; empirical runtime comparisons to baselines as dataset size and number of attributes increase.

- Organization: the organization of the "Method" part seems a bit disorganized. The authors should consider putting the algorithm in the main text or at least showing the overall algorithm first as a high-level overview and then breaking down each component in detail in subsections.

**Questions:**

- Is there any discussion/theoretical insights as to why the proposed method can perform better with limited sensitive information? More specifically, how does the fraction of samples $v$ play a part here?

- Are all experiments implemented with the same $v$ for different methods? Can authors also show how the performance (accuracy and fairness) change with $v$ for different methods? That may make the claim "sensitive-information-free" more convincing.

- Can authors also show the trade-off between accuracy and fairness for other methods to better compare?

- Is there more analysis in terms of running time/computational efficiency?

- How DP/Eod/Eop is measured in the experiments?

- What's the motivation behind choosing threshold $t$? How should we choose $t$?

- Is it possible to consider multiple sensitive attributes at the same time? In the experiment, the authors show two panels for race/gender respectively, if there are more than two sensitive variables, does it mean we need to train the model many times?

- The authors may consider using vector graphics to make the figures clearer.

---

### Official Review · Reviewer_kbuw · 2024-11-02

**Soundness:** 2
**Presentation:** 3
**Contribution:** 2
**Rating:** 5
**Confidence:** 4

**Summary:**

This paper proposes a two-stage debiasing framework for scenarios with limited demographic: (1) disentanglement of features using Nonlinear Independent Component Analysis (ICA) to separate task-relevant information (Y) from sensitive attributes (A), and (2) selective dimension reduction via Maximum Mean Discrepancy (MMD) ranking to remove dimensions highly correlated with sensitive information while preserving task performance. Empirical evaluations on three standard datasets (Adult UCI, CelebA, and UTKFace) demonstrate improvements over existing methods across multiple fairness metrics (Demographic Parity, Equalized Odds, and Equal Opportunity) while maintaining competitive accuracy.

**Strengths:**

- The use of Nonlinear ICA for fairness disentanglement is theoretically well-motivated, offering identifiability guarantees in contrast to common approaches like beta-VAE
- The approach maintains competitive accuracy while improving fairness metrics

**Weaknesses:**

The paper is not well defined regarding its theoretical objective. The authors repeatedly claim they can mitigate several fairness criteria simultaneously, without acknowledging their inherent incompatibilities. Specifically the claim "we can simultaneously improve all three fairness metrics" (p.9) overlooks that Demographic Parity (DP) and Equal Opportunity (EO) are often incompatible with each other [1].
While empirical improvements are shown for CelebA and UTK datasets (Figure 5), this success cannot be generalized, particularly for datasets with strong Y-A dependencies

The paper presents inconsistent formulations of the disentanglement function, varying between the equation, algorithm, and implementation (see my questions 2 to 5)

[1] Kleinberg, Jon, Sendhil Mullainathan, and Manish Raghavan. "Inherent trade-offs in the fair determination of risk scores." ITCS 2017

**Questions:**

1. Given that dimensions correlated with sensitive information S are removed from the latent space without conditioning on Y, it remains unclear whether your method is exclusively optimizing for Demographic Parity. Please clarify the theoretical optimality of your approach.
2. In Equation 2 of the main paper, the function $f^{⁻¹}$ takes x as its sole input. However, in Algorithm 1: Learning Disentangled Representations, both x and u are used as inputs (which differs from Algorithm 2 where it takes x only). Further, your code implementation shows x_train containing both x and u. Could you clarify this discrepancy in the input specification and explain if the inclusion of u alongside x is intentional?
3. In the original GIN paper [2], disentanglement by nonlinear ICA is formulated according to Equation 2 but omits the third term that corresponds to the log-determinant of the Jacobian. Why is this term needed in your approach when the transformation zhat is volume-preserving, as mentioned in [2]?
4. In your Adult dataset implementation, there appears to be an inconsistency with the target variable. In your implementation ('Adult/Step_1_Representation_Learning.ipynb'), the code extracts features 9-15 (x_train.iloc[:, 9:15]) which includes both relationship and workclass variables. The argmax operation on dimension 1 defaults to only consider the first variable (relationship), rendering the workclass variable ineffective. This conflicts with your appendix stating 'Adult dataset, we select the auxiliary variable u = Work class ∈ {Private, Self-emp, Gov, Without-pay}.' Could you clarify the intended target variable selection?"
5. there also appears to be a dataset size inconsistency, End of page 10 states 'the Group-labeled dataset has 3017 samples', however your code uses 7541 samples. Is this an error? The shared code uses test_size=0.25; it seems that test_size=0.1 would be needed to obtain 3017 samples. **I am unable to reproduce the experiments shared in your main paper.**

[2] Peter Sorrenson, Carsten Rother, Ullrich Köthe. Disentanglement by Nonlinear ICA with General Incompressible-flow Networks (GIN). ICLR 2020

---

### Official Review · Reviewer_bcmz · 2024-11-07

**Soundness:** 3
**Presentation:** 3
**Contribution:** 3
**Rating:** 6
**Confidence:** 3

**Summary:**

This paper proposes a method for achieving fairness via Nonlinear independent components analysis to generate disentangled representations of input data.

The method removes dimensions correlated with the sensitive attributes in the latent space, requiring only sufficient sensitive data to measure such a correlation.

**Strengths:**

1. This paper is well written and motivates the approach well within the related work. The proposed model is very straightforward and leverages well-understood tools for the purpose.

2. The approach is significant in a few regards: first, it reduces the reliance on sensitive demographic data for fairness mitigation. At the same time, it is stronger than related methods which 'infer' sensitive features without explicit availability https://arxiv.org/abs/2302.01385, https://dl.acm.org/doi/10.1145/3488560.3498493. Second, this 'semi-supervised' setting wrt sensitive labels means that users might collect higher-quality labels rather than often statistically-estimated race, gender etc labels.

3.   The model evaluates quite well. Although several methods are quite similar, the unique problem setting still makes this an interesting result. Furthermore, the authors present several qualitative evaluations, including visualizing the latent space manipulation (Fig 5)


Overall, for the interesting problem setting, I think this meets the threshold for acceptance.

**Weaknesses:**

1. A primary challenge is whether this method scales well to multiple criteria. Because the latent space is tied to the sensitive attribute target (e.g. age), it seems the model need be retrained under varying fairness criteria, rather than a post-processing method that doesn't require base model retraining, but only the instance reweights. This furthermore means that this is strictly an in-processing method that isn't amenable to varying base models. This might be very limiting in applications.

2. Nonlinear ICA seems challenging in terms of interpretability (specifically in the tabular domain, not image). While ICA and PCA can be interpretable within a few clean components, in practice this is an art. Gradient Boosting models seem more amenable to extracting rules or feature attribution wrt fairness.

**Questions:**

1. How might you address the flexibility of this method, to save retraining under multiple label-sets?

---

### Note · Authors · 2024-11-22

**Comment:**

We sincerely thank all the reviewers for their valuable suggestions and insightful comments. We will refine our work based on your feedback. Thank you once again for your time and dedication.

**Withdrawal Confirmation:**

I have read and agree with the venue's withdrawal policy on behalf of myself and my co-authors.